# NuClick: From Clicks in the Nuclei to Nuclear Boundaries

Mostafa Jahanifar[⋆2], Navid Alemi Koohbanani[⋆1,3], and Nasir Rajpoot[1,3]

[1]Department of Computer Science, University of Warwick, Coventry
[2]Department of Research & Development, NRP Co., Tehran, Iran
[3]Alan Turing Institute, London, UK

**Abstract.** Best performing nuclear segmentation methods are based on deep learning algorithms that require a large amount of annotated data. However, collecting annotations for nuclear segmentation is a very labor-intensive and time-consuming task. Thereby, providing a tool that can facilitate and speed up this procedure is of great interest. Here we propose a simple yet efficient framework based on convolutional neural networks, named NuClick, which can precisely segment nuclei boundaries by accepting a single point position (or click) inside each nucleus. Based on the clicked positions, inclusion and exclusion maps are generated which comprise of 2D Gaussian distributions centered on those positions. These maps serve as guiding signals for the network as they are concatenated to the input image. The inclusion map focuses on the desired nucleus while the exclusion map indicates neighboring nuclei and improve the results of segmentation in scenes with nuclei clutter. The NuClick not only facilitates collecting more annotation from unseen data but also leads to superior segmentation output for deep models. It is also worth mentioning that an instance segmentation model trained on NuClick generated labels was able to ranked 1[st] in LYON19 challenge.

**Keywords:** Interactive annotating · nuclei segmentation · instance segmentation · computational pathology

## 1 Introduction

Appearance and shape characteristics of nuclei in histology images are important markers for the diagnosis of cancer and predicting patient outcome [1]. To quantify these features, one should first determine the boundaries of the nuclei, which requires lots of time and effort to achieve manually. To this end, automatic segmentation methods play an important role in facilitating this task.

Since the emergence of deep learning (DL) methods and their superior performance over classical methods (feature-based), the need for annotated data has increased significantly. Data-dependency nature of DL methods still imposes a huge burden on the human for providing annotated data. Despite the labor intensive nature of annotating nuclei within histology images, several datasets have been provided for training deep networks [2,3,4]. The question here is: how we

---

⋆ These authors contributed equally to this work.

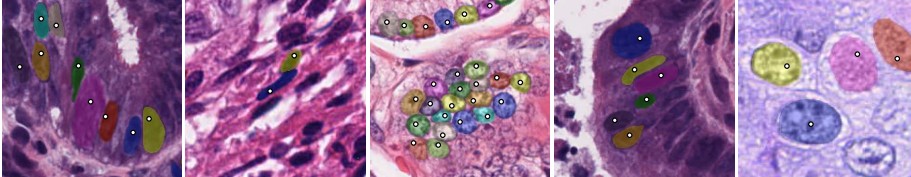

**Fig. 1.** Example outputs of NuClick: Annotator click inside the nucleus and the mask will be generated by NuClick model.

can use available annotated datasets to ease extending the knowledge and reducing the human effort when creating a new data set on another cancer/tissue type? In the computer vision domain, several methods have been employed to speed up the procedure of collecting annotations for natural images by accepting a few points from the annotator [5]. One of the most efficient models is DEXTR [5] which takes extreme points (the left-most, right-most, top-most and bottom-most pixels) of the object as the input to extract mask of the desired object.

All these approaches require the user to click on several points on the boundary of an object or draw a bounding box. For nuclear segmentation, providing several points on the boundaries of nuclei is still a high burden, since the annotator should first find the boundary of a nucleus in high magnification and then select several points on it. Moreover, nuclei are small objects, and their number may exceed 400 pixels in a patch size of 500×500 pixels (for example, when there is a dense cluster of lymphocytes), which makes this task more arduous. To the best of our knowledge, there is no similar approach based on DL models for interactive nuclei segmentation in histology images. Some works like [6] used the marker-controlled watershed algorithm to segment nuclei from marked points which failed in complex histology images.

Here, we propose a simple yet effective method for collecting nuclear annotation by asking a user to provide only one point inside the nucleus (examples are depicted in Fig. 1). Clicking one point inside an object is not a demanding task and can be done in low resolution by a non-expert. In summary, our contributions in this work are two-fold: 1) proposing a DL framework by adding two channels comprising guiding signals to the selected nucleus and its neighboring nuclei. 2) showing that the outputs from this framework can be useful in practice and for training deep networks.

## 2   Methodology

In the current work, we train the NuClick model for different labeled datasets of nuclei. For each data set, based on the centroids of annotated nuclei, patches are extracted from larger images, and then two guiding channels are created to serve alongside RGB patches as the network input. The network's parameters are then optimized based on a weighted hybrid loss function. On the other hand, during the prediction phase, our framework accepts an image and its marked nuclei (clicked positions) from the user as inputs and generate the instance segmenta-

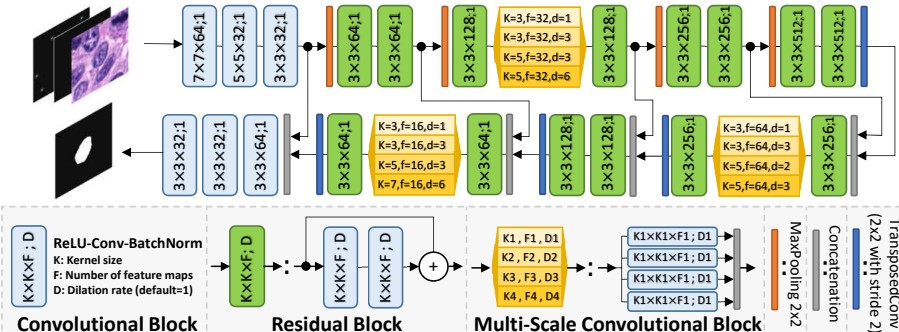

**Fig. 2.** NuClick network architecture. Comprising convolutional, residual, and multi-scale blocks. Level transition is done using MaxPooling and TransposedConv layers.

tion of the clicked nuclei in the output. In the rest of this paper, we describe each step in details.

### 2.1 Model architecture and loss function

We have utilized an encoder-decoder architecture, inspired by U-net [7], which reduces the size of feature maps in the encoding path while increasing the number of channels. The decoding path reverses this effect through several levels and turns those small and enriched feature maps into a single channel dense prediction. However, unlike the "traditional" U-Net, the NuClick architecture incorporates residual and multi-scale convolutional blocks [8] instead of normal convolutional layers in each level of encoding and decoding paths. An overview of the proposed NuClick architecture is depicted in Fig. 2. Using residual blocks enables us to train the network with higher learning rates without being worried about gradient vanishing effect [9]. Furthermore, multi-scale convolutional blocks allow the network to better capture the essence of image structures with different sizes and extract more relevant feature maps, hence boosting the network performance [8].

For training the network, we proposed to use a hybrid weighted loss function, which is based on a soft variant of the Dice similarity coefficient and weighted binary cross-entropy (1). The dice part of the loss controls class imbalance problem during training as most of pixels belong to the background, and weighted binary cross entropy penalizes the loss if network wrongly segments the neighbouring nuclei. Our proposed hybrid loss is as follow:

$$\mathcal{L} = 1 - \frac{\sum_i p_i g_i + \varepsilon}{\sum_i p_i + \sum_i g_i + \varepsilon} - \frac{1}{n} \sum_{i=1}^{n} w_i (g_i \log p_i + (1 - g_i) \log(1 - p_i)) \quad (1)$$

where $\varepsilon$ is a small number, $n$ is the number of pixels in the image spatial domain, $p_i$, $g_i$, and $w_i$ are values of the prediction map, the ground-truths mask, and the weight map at pixel $i$, respectively.

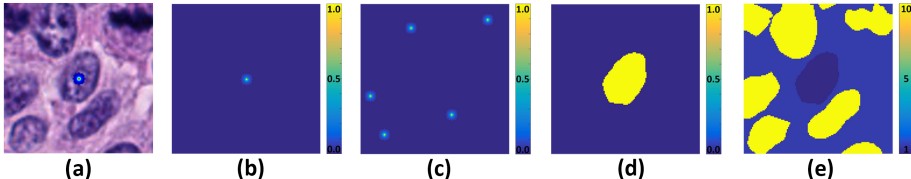

**Fig. 3.** Guiding signal maps: (a)-(c) show inputs to the NuClick network which are image patch, inclusion map, and exclusion map, respectively, (d) depicts the desired network output (ground truth), and (e) illustrates pixel-wise weight map used in the loss function.

The pixel-wise weight map is generated based on the ground-truth, where regions of the neighboring nuclei have 10 times more weight than the desired nucleus (marked object). To better understand these maps, a simple image patch with the desired nuclei clicked (marked) in it, alongside its related ground-truth and weight map are illustrated in Fig. 3. We incorporated this weighting scheme in the loss function to put more emphasis on the neighboring nuclei and avoid false segmentation of touching objects. In an alternative scenario, if we set the weights of the desired nuclei higher than other nuclei, the network may be falsely biased toward over-segmentation of neighboring nuclei.

### 2.2   Guiding signal maps

As a guiding signal and to incorporate prior knowledge, an extra channel is concatenated with the image in the network input which contains a 2D Gaussian distribution centered on the selected point (similar to [10] where the nucleus centroid was assumed as a Gaussian). We call this guiding signal the *inclusion map*, which refers to the nucleus we wanted to be included in the segmentation output. Adding the inclusion map helps the network to achieve a desirable segmentation of the selected nucleus as long as it is isolated. Based on our early experiments, although adding the inclusion map guide the model to segment the selected nucleus, in the presence of nuclei cluster the segmentation output might contain neighbouring nuclei too. To avoid this phenomenon and to exclude the neighboring nuclei in the output prediction map, we introduce the *exclusion map* as the fifth channel to the input which can contain multiple 2D Gaussian distributions centered on the clicked positions of the neighboring nuclei (if annotator provides them). For clarity, we display the inclusion and exclusion maps in Fig. 3(b)-(c) for a sample patch. They can also be seen at the input of the network in Fig. 2 as they are concatenated with the RGB image patch.

To this end, the inclusion channel always provides a guiding signal for segmenting the desired nucleus and if other nuclei in the vicinity of the patch are selected the exclusion mask will also provide a signal; otherwise it is an all-zero channel. Please note that within the training phase the inclusion and exclusion maps are generated on-the-fly, based on the augmented (changed) ground-truth mask, and during the test phase they are directly constructed based on the user clicked positions.

### 2.3   Training Procedure

To optimize the network weights, an Adam optimizer with initial learning rate of 0.003 was used. NuClick has been trained for 300 epoch with a batch size of 128 on all datasets. At each iteration, centroid positions of the desired nuclei are randomly jittered, and subsequently, the inclusion and exclusion maps are created on-the-fly. This makes the network more robust against the variations in the input position provided by the annotator.

### 2.4   Testing Procedure

During test time, for each input image, the user clicks on the nuclei for annotation, or the centroids are loaded from file. Afterward, for all available coordinates, patches of size 128×128 are extracted from image and inclusion, and exclusion maps are created as mentioned in the previous section. The NuClick predicts a nucleus segmentation map for each click (patch). Then that prediction map is converted to a binary map by thresholding, and objects with areas smaller than 10 pixels are removed (based on the size of the smallest object in the data set). The optimal threshold value, $T = 0.4$, is selected by testing a set of candidate values and evaluating the resulting binary maps. Moreover, for removing extra objects except for the desired nucleus inside the binary map, the morphological reconstruction operator is used which needs a marker and a mask. The marker has its all pixels equal to 0 except for a single pixel at the centroid location which is set to 1.The binary map plays the role of the mask in the morphological reconstruction. Having all patches predicted and processed, we can fill an empty canvas at the origin coordinates of each path with the processed nuclei masks to generate the final instance segmentation map of the input image.

## 3   Experiments and Results

### 3.1   Dataset

We have utilized two publicly available datasets in this work.First,the Kumar dataset [2] contains 30 images of size 1000×1000 which have been extracted from WSIs in The Tissue Genome Atlas (TCGA). This dataset contains seven tissue types and contains a total of 21623 nuclei instances segmented. From this dataset, 16 images are separated for training. The second dataset is CPM17 dataset [3] which consists of 32 images of size between 500×500 to 700×700 and a total of 7570 nuclei instances. Similarly, 16 images in CPM17 are used to extract patches as the training set.

### 3.2   Experimental results

To show the generalizability of the NuClick to an unseen dataset, Table 1 demonstrates quantitative results of NuClick when trained on a certain dataset (first column), and tested on another one (second column). Points for testing on the unseen dataset were provided from the centroids of objects in GT however they have been randomly jittered by 5 pixels to simulate manual annotation. We

have used six evaluation metrics: Aggregated Jaccard Index (AJI), general Dice similarity coefficient, object-wise Dice ($Dice_{Obj}$), Segmentation Quality (SQ), Detection Quality (DQ), and Panoptic Quality ($PQ=SQ\times DQ$) measures. The AJI metric [2] measures the quality of instance-wise predictions, DQ is equivalent to F1-score and only quantify the quality of detection, SQ reflects the average of intersection over union (IOU) for the detected object, Dice for evaluates the similarity of overall nuclei segmentation against the GT, and $Dice_{obj}$ measures the Dice coefficient for individual segmented nucleus. Comprehensive information about these metrics can be found in [11].

We also compare NuClick to two other approaches: U-Net, which is deep learning-based (supervised) model, and the watershed, which is an unsupervised method. For fair comparison in the case of U-Net, the detection map (Gaussian centered on each nucleus centroid) of nuclei is concatenated to RGB channels at the training and testing phases. Moreover, the watershed has been applied to the U-Net prediction to have instance-wise outputs. In the unsupervised framework, the marker-controlled watershed algorithm is applied to the gradient map of the image using the centroid points as the markers.

As reported in Table 1, the NuClick shows worse performance when it is trained on CPM and then tested on Kumar, which is due to some hard cases (Cancerous colon) in the Kumar dataset. Overall, NuClick performance, according to all metrics, is much better than the other two baselines that prove the high generalization capability of the NuClick. In an ideal situation, DQ for NuClick should be equal to 1, as it is representing the detection quality of the method and we have already provided the model with GT centroid defections. Nonetheless, DQ for NuClick is less than (yet very close to) 1. The reason is that NuClick does not consider some input points as valid nuclei, or the predicted map is eliminated during the thresholding and post-processing procedures. However, both detection and segmentation quality of NuClick is much higher than other reported methods in Table 1, which is also obvious in PQ metric. Quality of NuClick generated annotations is also evident from Fig. 1 which illustrates output masks for different clicked points in five image patches of different organs.

| Model | Train | Test | AJI | $Dice_{obj}$ | Dice | SQ | DQ | PQ |
|---|---|---|---|---|---|---|---|---|
| NuClick | CPM | Kumar | **0.7940** | **0.7937** | **0.8886** | **0.8001** | **0.9819** | **0.7856** |
| | Kumar | CPM | **0.8278** | **0.8278** | **0.9088** | **0.8361** | **0.9981** | **0.8180** |
| U-net+WS | CPM | Kumar | 0.7544 | 0.7601 | 0.8648 | 0.7823 | 0.9796 | 0.7663 |
| | Kumar | CPM | 0.7812 | 0.7844 | 0.8903 | 0.8074 | 0.9945 | 0.8029 |
| Watershed | - | Kumar | 0.1892 | 0.1660 | 0.4023 | 0.6936 | 0.3965 | 0.2805 |
| | - | CPM | 0.1501 | 0.1327 | 0.3467 | 0.7078 | 0.4243 | 0.3046 |

**Table 1.** Generalization of NuClick across CPM [3] and Kumar [2] datasets in comparison with other methods.

Moreover, to validate the quality of annotations generated by NuClick another experiment has been designed: we first train NuClick on CPM (Kumar) data, and then used the trained NuClick to generate labels for Kumar (CPM)

dataset. Afterwards, we train U-Net [7], FCN8 [12], and Segnet [13] models on NuClick's annotations for Kumar and CPM dataset. Performances of these models are compared against those of same models trained on GT annotations. Table 2 reports the results for this analysis. In this table, the title of each main column represents the name of the dataset that we apply our model on. Each sub-column of GT and NuClick$_{CPM/Kumar}$ indicate whether GT annotations or NuClick generated instances were utilized for training each model. Note that always GT annotations are used for model evaluation.

In Table 2, for all networks, we observe relatively similar results from outputs based on GT and NuClick annotations. For instance, when testing on Kumar dataset, Dice and PQ values from FCN8 model trained on NuClick$_{CPM}$'s annotations are 0.01 and 0.003 (insignificantly) higher than the model trained on GT annotations, respectively. This might be due to more uniformity of the NuClick generated annotations, which eliminate the negative effect of inter-annotator variations present in GT annotations. This example and negligible differences in metrics values for two scenarios in all cases, prove that labels provided by NuClick are good enough to train deep networks. Note that all hyper-parameters and the order of feeding patches during training are the same for all experiments.

| | Kumar | | | | CPM | | | |
| | GT | | NuClick$_{CPM}$ | | GT | | NuClick$_{Kumar}$ | |
| Models | **Dice** | **PQ** | **Dice** | **PQ** | **Dice** | **PQ** | **Dice** | **PQ** |
|---|---|---|---|---|---|---|---|---|
| U-net | 0.8243 | 0.5047 | 0.8196 | 0.5012 | 0.8535 | 0.5878 | 0.8458 | 0.5798 |
| Segnet | 0.8465 | 0.5238 | 0.8368 | 0.5178 | 0.8716 | 0.6268 | 0.8775 | 0.6281 |
| FCN8 | 0.7952 | 0.4484 | 0.8064 | 0.4512 | 0.8426 | 0.5998 | 0.8294 | 0.5904 |

**Table 2.** Comparative experiments on CPM [3] and Kumar [2] test set with models trained using GT and NuClick's predicted masks. NuClick subscript indicates the dataset that used for its training.

### 3.3   NuClick in Practice
**LYON19 Challenge** LYON19 is a scientific challenge on lymphocyte detection in immuno-histochemistry (IHC) sample images. Challenge organizers released a dataset comprising 441 images of IHC stained specimens of breast, colon, and prostate. The most challenging aspect of this task is that organizers did not release ground truth detection labels for the data set and instead asked the participants to use their data to develop a method. To develop a well-performing supervised method, particularly deep learning based models, annotated data is required. Therefore, NuClick was used to generate labeled data We transformed the centroid detection problem into a nuclei instance segmentation task, where for each image in the released data set, we randomly sampled a 256×256 patches to collect a subset of 441 training members. Then, a non-expert user reviewed all the patches and clicked on the positive lymphocytes based on his/her assumptions, which did not exceed 3 hours to be done completely. Image patches and their corresponding clicked positions are then fed into the NuClick framework to construct the instance segmentation map. After constructing a synthesized

ground truth for each image, we developed instance segmentation models for the LYON19 task. Extracted centroids from the output instances of our model were able to rank 1$^{st}$ in the LYON19 challenge leader-board achieving F1-score of 0.7951. This state-of-the-art result proves the fidelity of the NuClick generated masks once again and shows that NuClick can be used reliably in generating data sets for such tasks.

**PanNuke** Another use case of NuClick is in extending the dataset in our previous work, PanNuke [14], which demonstrates a pipeline for creating large classification and segmentation labels for nuclei. Here, we used NuClick to generate accurate nuclear segmentation masks, which was imperative when labeling thousands of nuclei.

## 4   Conclusion

We have proposed a simple and practical method for collecting nuclear annotation in histology images. We showed that one click from the user is enough to segment the nucleus, which is effortless and quick to collect a large number of annotations. Moreover, we have shown that the labels generated by NuClick are of high quality that can be used for training deep networks.

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
