# OpenReview forum: "NuClick: From Clicks in the Nuclei to Nuclear Boundaries"
_MICCAI.org/2019/Workshop/COMPAY — COMPAY 2019_

### Official Review · AnonReviewer2 · 2019-08-08
**Novel method for guided nuclei segmentation with UNET**

**Rating:** 7
**Confidence:** 3

**Review:**

SUMMARY OF MAIN FINDINGS
The best nuclei segmentation algorithms use deep learning methods which require extensive amounts of annotated data. While there are some data sets available, additional annotated data is needed to extend research. However, annotating data requires much labor. To reduce the workload needed to create nuclei annotations, the authors propose a tool that automatically produces nuclei annotations from manually labeled nuclei centers.
A modified U-Net architecture is utilized for segmentation. Instead of conventional convolutional layers, the architecture incorporates residual and multi-scale convolutional blocks. Furthermore, a hybrid loss function of dice similarity and weighted binary cross entropy is proposed for training.
A signal map, containing a 2D Gaussian distribution centered at the manually chosen point, is used to guide the U-Net architecture for segmentation. The map is concatenated with the image at the input and signifies the desired nuclei to be segmented. An additional 2D Gaussian exclusion map based on neighboring nuclei is included at the input to avoid overlapping/merged nuclei at the output.
The authors compared the performance of different methods (U-Net, Watershed, and the proposed, NuClick) that either used their generated masks or the original masks. The authors found that their segmentation scheme and mask generation methods produced comparable results to U-Net and Watershed segmentation methods.

DETALED COMMENTS
The strengths include:
1. The proposed method for nuclei mask generation is novel and innovative.
2. The proposed method addresses an important topic in digital pathology, that is, availability of annotated data.

---

### Official Review · AnonReviewer3 · 2019-08-12
**Practical tool to facilitate data annotation for cell nuclei segmentation algorithms**

**Rating:** 7
**Confidence:** 3

**Review:**

This paper focuses on facilitating the collection of annotated data to train deep-learning algorithms for cell nuclei segmentation. It presents a simple CNN-based approach named NuClick which is able to segment nuclei boundaries from a single point position inside each nucleus. The approach works by generating inclusion and exclusion maps based on the input positions which serve as guides for the network by concatenation to the input image. The results of experiments on two publicly available data sets as well as on the LYON19 challenge demonstrate the potential of the approach.

Overall the paper is clear and well written (safe for many typos as indicated below). The proposed idea is interesting and has high practical value. But the results raise some questions (see below) that need to be addressed.

Specific comments:

- Section 2.4: "...prediction map will be converted to mask by thresholding, and small objects are also removed." How is the "threshold" selected? What is the definition of "small"?

- Section 3.2, Table 2: It is unclear how in a few cases, training on NuClick (which itself is trained on another data set) can give better results than training on the ground truth (of the actual data set considered). This is very counter intuitive and needs explanation.

- Section 3.3: "...rank 1st in the LYON19 challenge leader-board with a high margin to other competitor." It is unclear what your F1-score was and thus how "high" the actual margin was. Please clarify.

Minor:

- Refer to Figure 1 in the main text.

- Improve English writing. Just some examples:

Page 2: "...adding two channel..."

Page 3: "...most of pixels are belonged to..."

Page 4: "...regions of the neighbouring nuclei has..."

Page 4: "...the inclusion channel always provide..."

Page 6: "...are a deep learning and unsupervised methods..."

Page 6: "...NuClick do not consider..."

Page 7: "Practise" -> "Practice"

Page 7: "...organizer did not released..."

Page 8: "...state-of-the-art results proves..."

---

### Official Review · AnonReviewer1 · 2019-08-12

**Rating:** 7
**Confidence:** 4

**Review:**

The authors present a semi-automated method to perform nuclei segmentation based on a modified U-net architecture. The main novelty of the paper lies in the addition of a cost-function and inclusion and exclusion maps added to the input to make sure adjacent cells do not interfere with the segmentation. I think the approach is interesting and could help quickly generate large sets of training data for nuclei segmentation in different projects. I do see some weaknesses with the paper:

- The postprocessing is very sparsely explained: how were the hyperparameters of the postprocessing optimized?
- It would have been interesting to see an ablation experiment where the contribution of the inclusion and exclusion maps was shown.
- The description for the results in the challenges is quite sparse: please use quantitative results as these also allow comparison over time to other methods.
- Some qualitative results on these datasets would have been good to include as well

---

### Decision · Program_Chairs · 2019-08-20

Accept